# Salt Preference and Ability to Discriminate between Salt Content of Two Commercially Available Products of Australian Primary Schoolchildren

**DOI:** 10.3390/nu11020388

**Published:** 2019-02-13

**Authors:** Madeline West, Djin Gie Liem, Alison Booth, Caryl Nowson, Carley Grimes

**Affiliations:** 1Institute for Physical Activity and Nutrition (IPAN), Deakin University, Locked Bag 20000, Waurn Ponds, Geelong, Victoria 3220, Australia; m.west@deakin.edu.au (M.W.); gie.liem@deakin.edu.au (D.G.L.); alison.booth@deakin.edu.au (A.B.); caryl.nowson@deakin.edu.au (C.N.); 2Centre for Advanced Sensory Science (CASS), School of Exercise and Nutrition Sciences, Deakin University, 1 Geringhap Street, Geelong 3220, Australia

**Keywords:** salt, taste, food preference, children

## Abstract

Australian children consume too much salt, primarily from processed foods where salt is often used to enhance flavour. Few studies have assessed children’s salt preference in commercially available foods. This study aims to assess (1) children’s preference and ability to discriminate between salt levels in two commercially available foods and (2) if preference or ability to discriminate between salt levels changes after an education program. Chips and corn flakes were tasted at three levels of salt concentration. Children ranked which they liked best (preference) and which was saltiest (ability to discriminate). The proportion of children across categorical responses was assessed (Chi squared and McNemar’s test) together with changes in preference and ability to discriminate between salt levels from timepoint 1 (T1) to timepoint 2 (T2). Ninety-two children (57% female, mean age 9.1 years (SD 0.8)) participated. At T1 approximately one-half and two-thirds of children preferred the highest salt chip and cornflake, respectively, (both *p* < 0.05). Fifty-seven percent and 63% of children identified the highest level of salt in chips and cornflakes as the saltiest, respectively. Preference and ability to discriminate between salt levels were unchanged between timepoints. Results support product reformulation to decrease salt content of foods provided to children.

## 1. Introduction

Children tend to prefer foods with higher levels of salt, with some evidence suggesting preferred levels of salt are higher in children than adults [1,2]. Notably, this high salt preference coexists with high intakes of salt among children and adults [3]. For example, in Australia, average salt intake among primary schoolchildren is 6.1 g/day [4] and among adults it is 9.6 g/day [5]. This exceeds the World Health Organization’s recommendation of no more than 5 g of salt per day [6]. The increased palatability of high salt foods may contribute to high salt intakes [7] as taste is an important predictor of food choice and consumption during childhood [8,9,10]. Therefore, if children develop preference for lower salt food products, this may lead to a lower daily salt intake. Whilst the link between dietary salt intake and preference for salty taste remains unclear in children [1,2,11], dietary habits have been found to track over time [12,13]. Thus, suggesting that choosing and consuming lower salt products during childhood may persist into adulthood. Reducing salt exposure during childhood is important due to links of high salt intake with raised blood pressure [14,15] and excess body weight [16,17].

Currently, there is wide variety in salt content of commercially available food products [18]. For example, Australian ready-to-eat breakfast cereals were found to vary in sodium (i.e., salt) content from 4 to 1063 mg/100 g [18]. Salt content alone cannot predict how salty a food will taste, as the location of salt within a food matrix and the type of matrix will influence salty taste perception [19]. If salt molecules are bound within the matrix, salty taste responses may not be elicited as the molecules will not be free to make contact with sodium dependent channels, which are responsible for the perception of salty taste [20]. Examples of foods that can be high in salt but may not taste salty are some cheeses and meats where salt is added for processing and maturation [21], and water-holding capacity [22], respectively. Furthermore, taste–taste interactions occur between salt and other taste profiles within the food matrix. Such interactions can result in suppression of bitterness and enhancement of sweetness [23]. Hence, the impact salt has on flavour is highly specific to the food it is added to [24], therefore salt preference appears to be food-specific [25,26,27].

In adults, experimental evidence shows that salt preference may be altered via changes to salt exposure through reduced salt diets and the subsequent increase in perception of saltiness [28] leading to a preference for lower salt foods [29,30,31]. Whether the same relationship is present in children is yet to be confirmed. A few cross-sectional studies have found inconsistent results [1,2,11], where salt preference in soup was not related to salt exposure as measured via a salt usage questionnaire [1]; salt preference in broth and crackers was found to be related to reported intake of salty foods [2]; and lastly, salt preference measured via one multiple choice question (children could answer dislike, no preference or like) was not related to salt intake measured via one spot urine [11]. The discrepancy in measures used in the above studies may account for inconsistencies in results.

Salty taste perception varies between individuals and is likely a function of exposure to dietary salt [29,30,31]. Children may perceive and detect salty tastes differently to adults [32], with higher amounts of salt in products needed to reach maximum preference ratings [1,2,33], as has been demonstrated in soups [33]. In adults, it has been found that those consuming high salt diets prefer higher salt foods compared to those on lower salt intakes as the taste sensory system adapts with a preference for high salt foods following repeated exposure [31]. Individuals then require greater amounts to achieve the most preferred level of salt in food [31]. This can be noted during intervention studies where dietary salt intake is altered by either increasing [31] or decreasing [29] the amount of salt added in the preparation of foods. Studies increasing salt intake through the use of salt capsules have not demonstrated changes in salt preference [34], highlighting that salt preference appears to be related to the amount of salt delivered to taste receptor cells in the mouth. No intervention studies targeting salt preference through total dietary salt intake have been completed in children and, hence, it is unknown whether perception or preference for salt in foods may be altered with manipulation of dietary salt intake.

Preference for high salt versions of home-cooked foods have been shown to influence intake, with children consuming more of preferred high salt pasta and green beans compared to lower salt options [25]. Research investigating children’s preference for and ability to detect differing levels of salt in commercially available Australian products is yet to be completed. As processed foods make such a large contribution (75–80%) to daily salt intake [35], understanding children’s preference for salt in these commercially available foods is important to help guide public health efforts which seek to reformulate processed foods to contain less added salt. The aims of this study were to assess salt preference for commercially available foods, e.g., potato chips and cornflakes, and if preference or ability to rank samples (ARS) according to salt content changed following an education program to reduce salt intake. It was hypothesized that children would prefer higher salt varieties of potato chips and cornflakes. Furthermore, following an education program to reduce salt intake, preference for high salt samples will decrease and children’s ability to rank samples (ARS) according to salt content will increase.

## 2. Methods

### 2.1. Study Design and Participants

Data for this analysis come from the Digital Education to LImit Salt Intake in the Home (DELISH) study. This was a single-arm study with a pre- and post-test design, and the methods have been previously reported [36]. Children in grades 2–4 (aged 7–10) were recruited from Government primary schools in the Greater Geelong region of Victoria, Australia. Schools were recruited from areas that varied in socio-economic background. For logistical reasons related to data collection, if schools had a low response rate (<10 children agreeing to participate), they were excluded [37]. To participate, the children required access to a home computer with Internet connection and a parent with email access. Children with food allergies were excluded.

The DELISH program was a 5-week Web-based salt reduction educational program, which aimed to reduce salt intake among children by 20%. Weekly online education sessions delivered to children targeted three key behaviours: (1) stop using the salt shaker; (2) switch to lower-salt foods by checking food labels (focus on bread, breakfast cereal and cheese); (3) swap processed salty foods (e.g., processed meats, take-out pizzas and burgers, savoury sauces and snack foods) with healthier alternatives. Parents also received supporting online newsletters. Further information about the intervention can be found in [31]. Ethics approval was given by Deakin University Human Ethics Advisory Group, Faculty of Health (HEAG–H 37_2016) and from the Victorian Department of Education and Training (2015_002884). Consent was obtained from participating school principals and classroom teachers, the primary caregiver of the child and assent from all children. Funding for this study was provided by the Heart Foundation (Vanguard Grant Application ID: 100574).

### 2.2. Data Collection

A team of trained research assistants travelled to each school for one day of data collection, where anthropometric and taste-testing data was collected. Children completing school day 24-h collections were assisted (research assistants were on hand to store and distribute materials), and those completing weekend collections were provided with materials and instructions to take home.

### 2.3. Demographic Characteristics

Parents completed demographic questionnaires at time point 1 (T1), providing basic parent and child demographic information, e.g., sex, date of birth and educational attainment, which was used to group participants into high, mid and low socio-economic status (SES) (high: university qualification, mid: completed high school, low: completed some high school) [38].

### 2.4. 24-h Urine Collection

Salt intake was assed via 1 × 24-h urine collection completed on either a school day or weekend day at time point 1 (T1) and again at time point 2 (T2). Twenty-four-hour urinary sodium excretion is considered the gold standard measurement for salt intake as 85–90% of ingested sodium is excreted in the urine [39]. Hence, a 24-h urine collection will capture salt ingested from all sources, including that added at the table. The urine collection protocol has been previously reported [36]. In brief, children were instructed to collect urine over any 24 h period that suited them, discounting the first urine and finishing the collection with one final urine. Urines were considered valid if the collection time was 20–28 h, total volume was >300 mls, creatinine was >0.1 mmol/d/Kg body weight and reported missed collections were <1 [36]. Based on the molecular weights of sodium (23 g/mol) and sodium chloride (58.5 g/mol), a conversion factor of 2.54 was used to convert sodium intake (g) to daily salt intake (g) [40].

### 2.5. Anthropometry

Standard protocols were followed for height and weight measurements and have been previously reported [36]. The International Obesity Taskforce body mass index cut-offs for children [41,42] were used to categorise children as underweight, healthy, overweight and obese.

### 2.6. Taste Testing

Two commercially available food products were tested—one snack food (potato chips, i.e., cold packaged crisps) and one staple food (corn flakes), both of which are key sources of dietary salt for children [43]. Study investigators sampled a range of chip and cornflake products of varying salt concentrations prior to selecting three samples of varying salt content for each food, i.e., low, medium, high concentrations (Table 1). Nutrient profiles, particularly sugar and fat, were matched as closely as possible between samples, as was visual appearance. On the data collection day, all product packages were kept out of children’s vision to ensure brands remained anonymous.

To ensure children understood the concept of saltiness before the taste testing session began, a research assistant presented a saltshaker to the child and explained that when salt is added to food it makes it taste salty. To check comprehension the child was then asked to give an example of a salty food.

Two taste tests were conducted during one session, (i) preference test and (ii) Ability to Rank Samples (ARS) according to salt content. The order of food presentation was randomised for every participant on two levels—food type presented first/second, and order of the three samples of varying salt content—for both tests. The two tests followed the same procedure, differing only in the question asked. Children were randomly presented with and tasted all three samples of one food at a time (potato chips or corn flakes), taking a sip of water between each sample. They were then asked, “Which one do you like the most?”, which represented preference. The identified sample was removed, and the child tasted the remaining two samples and was asked again, “Which one do you like the most?” Therefore, a rank order for the child’s preference was established. During the ARS test the child was asked, “Which one is the most salty?” This rank order method is based on previous methods [44] and has been successfully employed for use in a similar sample of children (mean age 7.4) when ranking sour intensities and preference [45].

### 2.7. Statistical Analysis

All statistical analyses were completed using STATA/SE-version 14.0 (StatCorp LP, College Station, TX, USA, 2015). Descriptive statistics (mean, SD or n, %) were used to describe characteristics of participants. Normality of data was assessed by visual inspection of histograms. Salt preference and ARS data were not normally distributed and non-parametric statistics were used. All analysis involving preference and ARS data used the top ranked sample, i.e., for preference this was the most preferred sample (1st preference), and for ARS this was the sample identified as the most salty. A *p*-value of <0.05 was considered significant. A Chi square goodness-of-fit test assessed the proportion of children across each of the three categorical responses (i.e., high, mid, low identified as most preferred or most salty). To assess change in preference and ARS from T1 to T2, participants were dichotomised based on the sample they most preferred at T1 (1—high salt sample most preferred, 2—low or mid salt sample most preferred). McNemar’s test assessed the change in the proportion of children across the two categories from T1 to T2. The same method of grouping and analysis was used to assess changes in ARS.

## 3. Results

### 3.1. Demographic Characteristics

One-hundred and two students consented to participate in this study. Two students dropped out prior to T1 measures due to absence from school and 8 children were excluded (*n* = 2 due to low consent and exclusion of school, *n* = 2 due to food allergies, *n* = 4 were missing parent consent for taste testing session). Dropouts after T1 measures were (*n* = 15) for a range of reasons listed in Figure 1. Additionally, (*n* = 1) was excluded from T2 measures as T1 data was incomplete due to the student leaving school early to attend an excursion. Therefore, at T1 the sample consisted of 92 children, with over half being female (56%). The average age of participants was nine years and the majority were of a healthy weight (Table 2). Findings related to change in salt intake from T1 to T2 have previously been reported [37]. Salt intake remained unchanged at T2 (*n* = 51 children with a complete 24-h urine collection at both time points: T1 5.4 (SE 0.4) g/day vs. T2 5.3 (SE 0.3) g/day, *p* = 0.75).

### 3.2. Salt Preference

The distribution of children’s preference for potato chips was not evenly spread across the three categories with nearly half of the children preferring the chip with the highest salt concentration (*p* = 0.02) at T1. At T2, a similar proportion preferred the highest salt chip; however, the uneven distribution was no longer statistically significant (*p* = 0.07) (Table 3). At T1, approximately two-thirds of the children preferred the highest salt corn flake (*p* < 0.01), with similar results at T2 (*p* < 0.01) (Table 3).

### 3.3. Ability to Rank Samples (ARS) According to Salt Content

Over half of the children (57%) correctly identified the high salt potato chip and the high salt corn flake as tasting the saltiest (63%) at T1 (Table 4). Results remained similar at T2 with 49% and 62% of children correctly identifying the highest salt potato chip and corn flake, respectively. Less than 6% and 17% incorrectly identified the low/no-salt potato chip and cornflake as having the highest salt content, respectively.

### 3.4. Changes in Salt Preference from Time Point (T1) to Time Point 2 (T2)

Of children with taste data for both time points (*n* = 76 for potato chips and *n* = 74 for cornflakes), there was no significant difference in the proportion who changed their top preference from high salt potato chip to a lower salt potato chip (*n* = 18, 24%), or vice versa, e.g., changed from a lower salt potato chip to a high salt potato chip (*n* = 18, 24%) from T1 to T2 (*p* > 0.99) (Figure 2). Furthermore, the most preferred potato chip did not change for 53% (*n* = 55) of children following the education program (i.e., these children preferred the same salt content at T1 and T2). Similarly, there was no statistically significant difference in the proportion of children who changed their most preferred corn flake from high salt to low salt (*n* = 12, 16%) or vice versa (*n* = 7, 9%) (*p* = 0.36) (Figure 2). Seventy-five percent of children remained preferring the same sample of cornflakes at T2 as T1 (Figure 2).

### 3.5. Changes in Ability to Rank Samples (ARS) according to Salt Content from T1 to T2

At T1 (*n* = 42) 55% of children correctly identified the highest salt chip as the saltiest and the rest (*n* = 34, 45%) incorrectly identified the lower salt chips (i.e., low or medium) as the saltiest (Figure 3). Similar to preference data, the observed shift of children who identified the alternative option (i.e., went from correctly identifying high salt chip at T1 to incorrectly identifying lower salt chips at T2) (*n* = 25, 33%) or vice versa (*n* = 20, 26%), was random with no significant trend to indicate that ARS changed during the intervention. The same finding was also found for cornflakes, with no improvement in children’s ability to correctly identify the highest salt option following the intervention.

## 4. Discussion

This study provides the first insight into Australian children’s preferences for and ability to discriminate between salt content in two commercially available foods: potato chips (sodium content 14–489 mg/100 g) and cornflakes (sodium content 90–590 mg/100 g) before and after an education program to reduce salt intake. Overall, in this sample of children, participation in a 5-week salt education program did not change salt intake (i.e., 5.4 vs. 5.3 g/day at T1 and T2, *p* = 0.75) [37], and preference and ARS remained unchanged. The key findings indicate that over half the children (49–57%) were able to correctly identify the highest salt potato chips and corn flakes (62–63%), and that these most salty brands are preferred by 45–47% and 55–61%, respectively, for each food. However, the lack of consistency within individual children in correctly identifying the highest salt chip and cornflake at the two time points raises some concerns: only 22% correctly identified the highest salt chip on two occasions, whereas 43% achieved consistency between the two time points in identifying the highest salt cornflake. It is not clear if some other property of the cornflake, e.g., colour or texture alerted children to the saltiest sample or if it was easier for the children to identify the saltiest cornflake. With respect to preference, 75% of children were consistent with their preference for cornflakes between the two time points, with only 25% changing preference, although for potato chips preference remained the same for only 53% of the children with 48% changing. This might suggest that a proportion of the children were not able to discriminate between the different salt levels in chips, rather than children not understanding the preference task. It should be noted that the sugar levels in potato chips, unlike the sugar levels in corn flakes, were similar across the three different types of potato chips. Alternatively, but not mutually exclusive, children may have picked up the difference in sugar levels between the three different cornflakes or the interaction between sodium and sugar, rather than the difference in sodium levels per se. This could explain why some children did not pick up the different sodium levels in chips. However, this remains to be investigated. The current study did not assess children’s sweetness perception of any of the products.

Children’s preferences for foods high in salt have been documented throughout the literature [1,2], therefore it was expected that high salt samples of potato chips and cornflakes would be the most preferred. Drawing on our knowledge of the relationship in adults, if salt intake did not change, we would expect salt preference and ARS to remain unchanged. This is due to the underlying mechanisms of altering salt preference, which relies on altering exposure to salty foods. Studies in adults have shown low-salt diet interventions of 5 and 12 months result in decreased salt preferences when measured in solution, soup, crackers [29] and cream of green soup [30], respectively. In these studies, changes in preferences were seen between several to 13 weeks after initiating the low-salt diet. This suggests that even if the children were to have decreased their salt intake, just 5 weeks consuming less salt may not have been sufficient to have any impact on salt preference and that any reductions in salt intake would have been very modest.

Of note, only one of the taste-tested foods was specifically targeted in the education program. One week, the focus of the program was to switch to a lower salt breakfast cereal or bread. If children were successful in this task, repeated exposure to lower salt cereals could potentially have induced a preference for such lower salt cereals [46].

The presented preferences for higher salt varieties of commercially available Australian foods highlights the need for the implementation of sodium reduction targets on processed foods to encourage food manufactures to add less salt during processing. Gradual step-wise reductions in salt content have been shown to be possible without consumer’s detection [47]. This approach to reducing salt content in the food supply has been successful in the United Kingdom [48] and is currently being advocated for in the United States [49], and may prove to be more effective in reducing salt intake than individuals having to choose lower salt products for themselves.

### Strengths and Limitations

The salt preference measurement used in this study is both a strength and a limitation of the design. The forced-choice rank method has previously been shown to be suitable for use in preference tests with children [44,45]. However, discrepancies in this measurement have been highlighted in this study, raising the question of whether this is a reliable measure of salt preference in children. Even though in the overall sample there was no significant change in children’s salt preference and ARS from time point 1 to time point 2, a large proportion of children changed the sample they identified as most preferred and the saltiest. From time point 1 to time point 2, 48% and 25% of children changed the potato chip and cornflake sample they identified as most preferred, respectively. Additionally, 59% and 39% of children changed the potato chip and corn flake sample they identified as most salty, respectively. The large number of random changes between preference and ARS warrants further investigation into the reliability of this measurement tool.

Furthermore, the choice of food used for measuring salt preference is also important. Previous studies have measured children’s salt preference in broths and soups [1,2] and salted vegetables [25], where salt is added to enhance flavour and represents salt added at the table or in cooking. Intake from such sources only accounts for 10–15% of overall salt intake [35]. Alternatively, salt intake in the form of “hidden salt” in processed foods accounts for 75–80% of total salt intake [35]. Few studies have recognised the need to measure salt preference in more real-world situations and therefore also used crackers in salt preference tests [2]. Measuring salt preference in commercially available foods is a major strength of this study and contributes to the novelty of the study. Results provide valuable information relevant to the current food market and may assist to inform future public health initiatives that aim to reduce salt intake in children.

Matching nutrient profiles was a priority for both potato chips and corn flakes; however, this was more difficult for corn flakes and resulted in a wider range of sugar content across the samples (high salt sample total sugar content= 8.6 g/100 g, mid salt sample = 4.9g/100 g, low salt sample = 6.0 g/100 g). Although the low salt sample had a higher sugar content than the mid salt sample, low salt corn flakes were visibly different, and the texture was thick and crunchy rather than light and crisp. These sensory properties may have influenced the proportion of children (*n* = 2) selecting these as the most preferred sample. Hence, even though using commercially available products is a strength of the study design, the inability to account for some key sensory differences is also a limitation.

Generally, there is wide variation in the sodium content of foods that fall within the same food category. For example, Webster et al., collated data on 80 potato chips, representing 99% of the Australian market and found that products ranged from 30 to 1404 mg sodium/100 g (mean 641/100 g) [13]. It is evident that the sodium content of chips used for this study (14 mg/100 g, 200 mg/100 g and 486 mg/100 g) are not representative of the whole market, which includes flavoured varieties of chips that can be much higher in salt than plain potato chips. Similarly, the range of sodium in ready to eat cereals was reported as 4–1063 mg/100 g [18], which is also not reflected in the chosen corn flake samples used within the current study (90 mg/100 g, 390 mg/100 g and 590 mg/100 g). For both foods, samples were chosen to represent the spread of sodium content evident in products available at supermarkets in the local area. Hence, the chosen sodium ranges reflect the spread in sodium content available to the general population for these foods in this particular geographical location. The exception being low salt potato chips, which needed to be sourced online as those available at supermarkets were much higher in fat, compared to mid and high salt chip samples, suggesting this particular sample may not be easily accessible to the population.

## 5. Conclusions

This is the first study detailing Australian children’s preferences for commercially available products of varying salt content. Furthermore, this is the first study to investigate whether children can differentiate between products with varying salt content. It is clear that children prefer commercially available products with higher salt, and at the group level, children can generally identify the highest salt version of potato chips and cornflakes available in the market. However, children’s preference and ability to detect salt in these tested foods remained unchanged following participation in a 5-week education program designed to lower salt in the diet. It is possible that to achieve reductions in children’s salt intake and preference for salt in foods, wider and more intensive strategies are required. Application of school based salt reduction education programs are likely more beneficial for raising awareness of high salt intakes in children, and for improving salt related knowledge, attitudes and behaviours of parents [50]. A reduction in the salt content of foods commonly eaten by children is a priority for health, and we know that gradual, step-wise reductions in the salt content of commercially produced foods is achievable without detection. The development and marketing of lower salt manufactured foods is necessary to ensure that children are exposed to commercial products with lower salt content.

## Figures and Tables

**Figure 1 nutrients-11-00388-f001:**
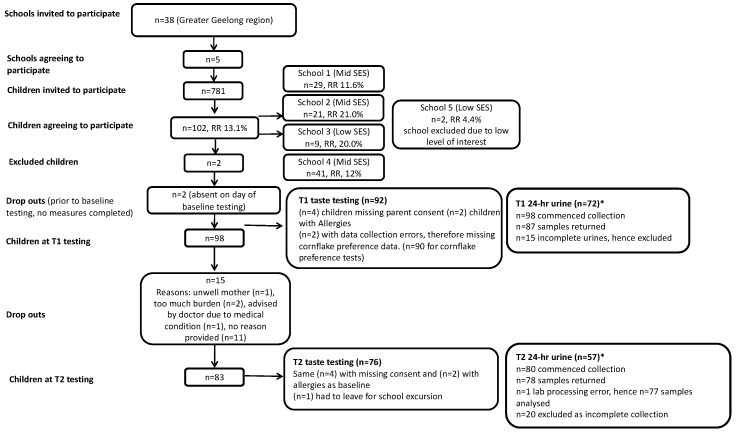
Flowchart of participant recruitment and retention within the Digital Education to Limit Salt in the Home program. * Further information on urine collection numbers have been previously reported [37].

**Figure 2 nutrients-11-00388-f002:**
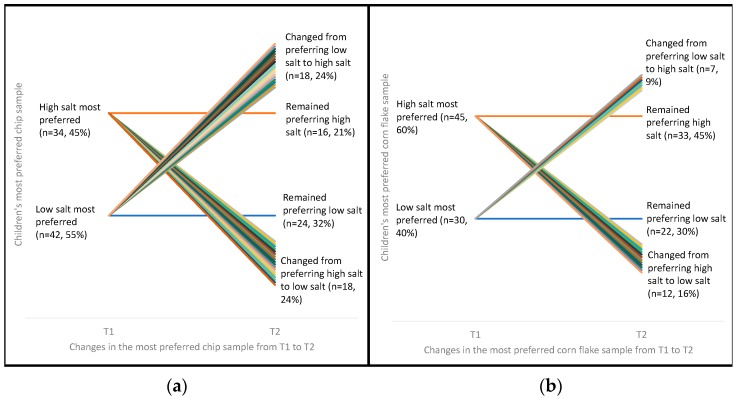
Participants’ most preferred potato chip (**a**) and cornflake (**b**) samples at T1 and T2 ^1,2,3,4,5^; ^1^ Chip McNemar’s Test, *p* > 0.99; ^2^ Cornflake McNemar’s Test, *p* = 0.36; ^3^ “Low salt” group includes both low and mid salt samples; ^4^
*n* = 2 participants missing from corn flake graph due to recording errors at data collection. *n* = 1 missing data for T1 and *n* = 1 missing data for T2; ^5^ The ‘change’ lines are a visual representation of the number of children who changed their preference from T1 to T2—each color is a different child. As the graph is showing the changes in preference over time, the ‘remain’ lines stay the same thickness at both times points—this thickness does not represent the proportion of children who remained preferring the same sample.

**Figure 3 nutrients-11-00388-f003:**
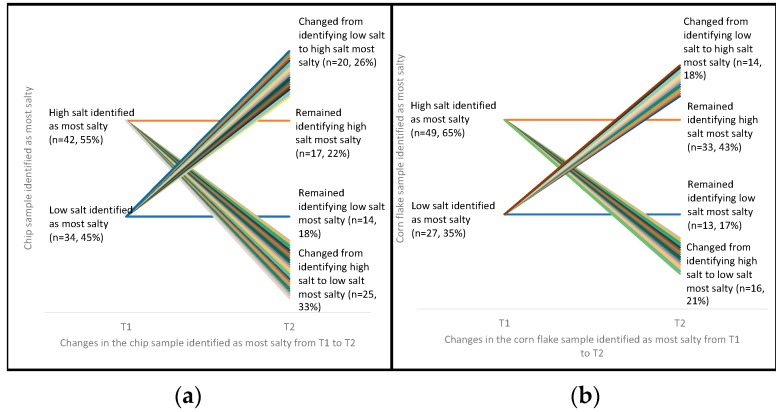
Participants’ identification of high salt potato chip (**a**) and cornflake (**b**) samples at T1 and T2 ^1,2,3,4^; ^1^ Chip McNemar’s Test, *p* = 0.55; ^2^ Cornflake McNemar’s Test, *p* > 0.99; ^3^ ”Low salt” group includes both low and mid salt samples; ^4^ The ‘change’ lines are a visual representation of the number of children who changed which sample they ranked as most salty from T1 to T2—each color is a different child. As the graph is showing the changes in ARS over time, the ‘remain’ lines stay the same thickness at both times points—this thickness does not represent the proportion of children who remained identifying the same sample as the most salty.

**Table 1 nutrients-11-00388-t001:** Sodium (i.e., salt) content of potato chip and corn flake samples (per 100 g).

Food Item	Sodium (mg)	Salt Equiv. (g)	Energy (kJ)	Sugars (g)	Total Fat (g)
**Potato chips**					
No added salt	14.0	0.04	2120.0	0.3	24.4
Mid salt	200.0	0.5	2200.0	0.3	30.0
High salt	486.0	1.2	2090.0	0.0	27.0
**Corn flakes**					
Low salt	90.0	0.2	1510.0	6.0	2.0
Mid salt	390.0	1.0	1580.0	4.9	0.7
High salt	590.0	1.5	1520.0	8.6	0.3

**Table 2 nutrients-11-00388-t002:** Characteristics of participating children at T1 data collection (*n* = 92).

	*N*	Proportion (%)	Mean	SD
**Sex**				
Male	40	44		
Female	52	56		
**Age**			9.09	0.78
**BMI category ***				
Healthy weight	73	79		
Overweight	12	13		
Obese	7	8		
**SES ****				
Low	21	26		
Mid	26	32		
High	35	43		
**Sodium intake (mmol) *****			91	41
**Salt intake (g) *****			5.35	2.43

* Based on the International Obesity Task Force BMI (body mass index) reference cut-offs [41,42]. ** *n* = 20 missing SES. *** As measured by 24-h urinary sodium excretion. Only includes participants with valid urines (*n* = 72).

**Table 3 nutrients-11-00388-t003:** Children’s most preferred samples.

	Potato Chips				Corn Flakes			
Salt content	T1 (*n* = 92)		T2 (*n* = 76)		T1 (*n* = 90) *		T2 (*n* = 76)	
	*n*	%	*n*	%	*n*	%	*n*	%
**Low/no salt**	22	24	18	24	3	3	15	20
**Mid salt**	27	29	24	32	32	36	19	25
**High salt**	43 ^1^	47	34 ^2^	45	55 ^3^	61	41 ^4^	55

* Cornflake preference data missing *n* = 2 students due to recording errors during data collection ^1^ Chi Square Goodness of Fit value (7.85), *p* = 0.02; ^2^ Chi Square Goodness of Fit value (5.16), *p* = 0.07; ^3^ Chi Square Goodness of Fit value (45.27), *p* < 0.01; ^4^ Chi Square Goodness of Fit value (15.68), *p* < 0.01.

**Table 4 nutrients-11-00388-t004:** Children’s identification of the most salty chip and cornflake sample at time points 1 and 2.

	Potato Chips				Corn Flakes			
Salt content	T1 (*n* = 92)		T2 (*n* = 76)		T1 (*n* = 92)		T2 (*n* = 76)	
	*n*	%	*n*	%	*n*	%	*n*	%
**Low/no salt**	2	2	4	5	15	16	12	16
**Mid salt**	38	41	35	46	19	21	17	22
**High salt**	52 ^1^	57	37 ^2^	49	58 ^3^	63	47 ^4^	62

^1^ Chi Square Goodness of Fit value (43.4), *p* < 0.01; ^2^ Chi Square Goodness of Fit value (27.03), *p* < 0.01; ^3^ Chi Square Goodness of Fit value (36.81), *p* < 0.01; ^4^ Chi Square Goodness of Fit value (28.29), *p* < 0.01.

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
