# Peer review of "Salt Preference and Ability to Discriminate between Salt Content of Two Commercially Available Products of Australian Primary Schoolchildren"

_nutrients, 2019, doi:10.3390/nu11020388_

Round 1

Reviewer 1 Report

It has been a pleasure to review this paper. This is a well written and clear manuscript

which analyzes the relationship between the sodium content of processed foods and the individual preferences of children when choosing food as well as their ability to discern between those foods that are more or less salty. It also evaluates whether an educational programme aimed at reducing salt consumption is capable of modifying their preferences and the ability to discern salt content.

I think there are some issues the authors should consider to improve the manuscript.

Title: The title should be revised, it is too brief, it does not include the main topics studied in this paper.

Material and methods:  It would be appropriate to include in this paper more information in the methods section in order to improve its reading. It was confusing to have to consult another manuscript to know all the methodology. For example:

- It should incorporated a flowchart of participants throughout the study.

- In relation to the 24-hour urinary sodium, it should be explained why it represents salt intake, how it is done the conversion of measured sodium in urine samples to sodium intake (change of units) and which methods were used to validate urine samples.

- It was not clear what cut-off points have been used to categorize children according to their weight. Have you used the cut-off points for the BMI? Why did you measure the height if only use weight? Why weight categories have been calculated and not the prevalence of overweight and obesity?

- It should be explained why cereals or potatoes were used instead of other processed foods.

Results and discussion:

- In table 2, add that sodium and salt are by 24 hours. If creatinine was used as a method to validate urine samples, the creatinine values should be in Table 2.

- Table 3: It would be interesting to include the values of sodium excreted in 24h urine 

samples according to children's preference for more or less salty foods and their ability to recognize the saltiest foods, according to the information given at the introduction.

- Figure 1 and 2: Figure 1 and 2 are difficult to interpret and it is suggested re-draw the Figures. Given the width and colours of the lines, it seems that more children change their preference than those who keep it the same between T1 and T2. When you look at the percentages, a large part of the children keep the first option they chose. Also, the same terminology should be used as throughout the writing. T1 instead of baseline and T2 instead of post-intervention.   

Conclusion: the conclusion could better conclude your whole research, answering to the proposed objectives.

Author Response

Please see attached word document.

Reviewer 2 Report

1.                   Authors are suggested to take care of spelling and grammatical errors.

2.                  In methods section urine collection has been done at both time points and assessed for what? What components were measured? What was the method? Where is the data for that? Pls provide the information in detail.

3.                  In introduction line 27 high salt intake was mentioned what is the normal salt intake/day both in adult and children. Please mention those ranges also

4.                  In introduction line 34 high salt intake not only raises blood pressure…… Please mention further complications also 

5.                  Implementation of diet (beneficial and adverse effects) studies in educational programs of schools might be better choice to provide the children awareness. Pls include  couple of sentences about awareness programs also  in conclusion.

Author Response

Please see attached word doc.
